# Inversion of Phytoplankton Pigment Vertical Profiles from Satellite Data Using Machine Learning

Agathe Puissant [1], Roy El Hourany [2,*], Anastase Alexandre Charantonis [1,3], Chris Bowler [2] and Sylvie Thiria [1,4]

1. Laboratoire d'Océanographie et du Climat Expérimentals et Approches Numériques (LOCEAN), Sorbonne Université, CNRS, IRD, MNHN, 75005 Paris, France; agathe.puissant@locean-ipsl.upmc.fr (A.P.); anastase.charantonis@ensiie.fr (A.A.C.); sylvie.thiria@locean-ipsl.upmc.fr (S.T.)
2. Institut de Biologie de l'École Normale Supérieure (IBENS), École Normale Supérieure, CNRS, INSERM, PSL Université, 75005 Paris, France; cbowler@biologie.ens.fr
3. École Nationale Supérieure d'Informatique pour l'Industrie et l'Entreprise (ENSIIE), 91000 Évry, France
4. Observatoire de Versailles Saint-Quentin-en-Yvelins (OVSQ), Versailles Saint-Quentin-en-Yvelines University, 78280 Guyancourt, France
* Correspondence: elhouran@biologie.ens.fr

**Abstract:** Observing the vertical dynamic of phytoplankton in the water column is essential to understand the evolution of the ocean primary productivity under climate change and the efficiency of the $CO_2$ biological pump. This is usually made through in-situ measurements. In this paper, we propose a machine learning methodology to infer the vertical distribution of phytoplankton pigments from surface satellite observations, allowing their global estimation with a high spatial and temporal resolution. After imputing missing values through iterative completion Self-Organizing Maps, smoothing and reducing the vertical distributions through principal component analysis, we used a Self-Organizing Map to cluster the reduced profiles with satellite observations. These referent vector clusters were then used to invert the vertical profiles of phytoplankton pigments. The methodology was trained and validated on the MAREDAT dataset and tested on the Tara Oceans dataset. The different regression coefficients $R^2$ between observed and estimated vertical profiles of pigment concentration are, on average, greater than 0.7. We could expect to monitor the vertical distribution of phytoplankton types in the global ocean.

**Keywords:** machine learning; inversion; ocean colour; phytoplankton; pigment vertical profile; deep chlorophyll maximum; Tara Oceans; MAREDAT; pigments; ITCOMP-SOM; Self Organizing Maps

## 1. Introduction

Phytoplankton is a key player in ocean biodiversity with consequences on fish catch potential, and climate regulation through carbon dioxide storage [1–4]. A decline in total phytoplankton population has been observed in Northern hemisphere basins over the last decade [5] and is projected to strengthen over the 21st century over wide oceanic regions under all global warming scenarios [6]. This decline is one of the most alarming consequences of anthropogenic climate change, as highlighted by recent policy-relevant reports [7] and by a scientists' warning to a humanity consensus statement in Nature Reviews [8]. However, an important question is how phytoplankton composition responds to changes in ocean characteristics (temperature, nutrients, currents, stratification, ...) since phytoplankton diversity constrains the societal impacts on both climate and fisheries.

Methods to observe the phytoplankton diversity from remote sensing data have greatly progressed during the last two decades [9,10]. New algorithms have been developed [11,12] that extract phytoplankton pigments and phytoplankton Functional Types (PFTs) at sea surface from satellite ocean color data. A major limitation of ocean color observations is that they only provide information on the sea-surface and miss subsurface peaks of phytoplankton abundance that can represent a large proportion of the total depth-integrated quantity. In fact, Morel and Berthon [13] classified the vertical variability into

"trophic categories" following the surface Chlorophyll-A (Chla) concentration, and showed that there is a relationship between this concentration and the integrated concentration of Chla in the water column. Subsequently, based on this previous work, Uitz et al. [14] determined from surface satellite data the variability of different phytoplankton size classes (PSC) in the water column based on their contribution to the Chla. However, these studies are constrained by the empirical relationships between Chla and secondary pigments and by assumptions on the shape of the vertical pigments profiles and cannot predict atypical associations [15]. Charantonis et al. [16] presented a combined use of a Self-Organizing Map with the Hidden Markov Models to infer Three-Dimensional Chla fields starting from Two-Dimensional (2D) imaging of several variables (surface Chla, Sea Surface Elevation, solar radiation and wind). Furthermore, Cortivo et al. [17] proposed a neural network methodology to estimate the sub-surface Chla concentration in open waters from the upwelling radiation. A similar attempt to infer the vertical Chla profile, by using a Multi-Layer-Perceptron (MLP), was shown in Sauzède et al. [18], in which the output is predicted from surface ocean-color estimates and depth-resolved physical properties, derived profiling floats such as SST and salinity. In addition, finally, Sammartino et al. [19] and Sammartino et al. ( [20] proposed a regional neural network approach to reconstruct the 3D variability of Chla in the Mediterranean sea. All of these works have targeted the Chla reconstruction as the main proxy of phytoplankton biomass. However, Uitz et al. [14] and Sauzède et al. [18] pushed their approach one step further to reconstruct phytoplankton community structure in terms of cell size.

In the present work, we introduce a new machine learning (ML) methodology to estimate several phytoplankton pigment profiles from ocean-color data, hindering a multi-dimensional problem based on the co-estimation of six different pigments. The novelty of this work lies within the ability of observing the 3D variability of phytoplankton functional types using these pigments.

Indeed, recent developments in artificial intelligence, combined with the availability of large datasets of satellite observations, provide enormous potential to learn the hidden structure of geophysical phenomena such as the one faced in this paper. ML methods have started to allow the intelligent investigation of such multi-dimensional data sets in oceanography and biogeochemical studies [21–23]. ML algorithms are now used to exploit spatial and temporal complex data structures, find patterns, and fuse heterogeneous sources of information efficiently. The survey in Reichstein et al. [24] describes the recent achievements and research challenges in the field of geophysics. Cross-fertilization of the ML with physical and biogeochemical contexts should allow the extraction of relevant knowledge from the dataset encountered in this study. This functioning is crucial for a better joint exploitation of observational data for understanding the phytoplankton variability as observed from space.

To achieve this aim, we used a large global database of pigment concentrations measured by high-performance liquid chromatography (HPLC) at the surface and through the water column, the Marine Ecosystem Data (MAREDAT) database [25], alongside with satellite ocean colour daily matchups. After a series of training and validation experiments on MAREDAT, we will use, as a final test, the HPLC data provided by Tara Oceans Expedition [26], a pan-oceanic expedition that deployed a holistic sampling of phytoplankton communities, coupled with comprehensive in situ biogeochemical measurements which provide the detailed environmental contexts necessary for ecological interpretation of the phytoplankton ecosystem.

## 2. Materials and Methods

### 2.1. Data

This section is devoted to the data we used that can be split into two distinct parts: in-situ observations and remotely sensed signals. Remote sensed data are abundant and easy to acquire, but the in-situ observations that are gathered during oceanic campaigns all around the world are sparse and represent a limited dataset. Due to the difficulty inherent

to measurements at sea, the in-situ dataset is heterogeneously sampled in both pigments and depths. Moreover, both datasets are imperfect and have a percentage of missing data that can be consequent. The challenge is thus to gather the available information (in-situ and remotely sensed) and to build a limited but robust dataset allowing the use of machine learning techniques. This requires the fusion of the two datasets.

### 2.1.1. Pigment Observations

The MAREDAT database contains concentration measurements obtained at different depths and different stations at sea and analysed by HPLC for Chla and secondary pigments. The stations, defined by their longitude, latitude, and date (day/month/year), come from 136 scientific cruises around the world which have been compiled and quality controlled [25].

Besides the Chla concentration, we used 5 pigments that provide information on the main groups of phytoplankton: Divinyl-Chlorophyll-A (DVchla), 19'hexanoyloxyfucoxanthin (19hex), fucoxanthin (fucox), peridinin (perid) and zeaxanthin (zeax). These pigments were chosen based on their ability to distinguish the main groups of phytoplankton determined from the scientific literature [14,27–29]; Fucoxanthin for diatoms [30], Peridinin for dinoflagellates [30,31], 19'Hexanoyl-Fucoxanthin for Haptophytes [32], Zeaxanthin for Cyanobacteria [33,34] and Divinyl Chlorophyll-a for Prochlorococcus [33,34].

The measurements corresponding to depths greater than 300 m have been eliminated due to low pigment concentration and variability in light-limited environments. A quality control check was performed to filter the data, described in the following paragraph.

First, measurements with Chla concentrations greater than $3\,\mathrm{mg\,m^{-3}}$ were rejected, as they correspond to rare and abnormal high concentrations encountered in open waters [11]. Afterwards, values of secondary pigments above the 95th percentile for each pigment were considered outliers and were replaced by missing values [11,29]. In addition, finally, due to specific physical, optical and biogeochemical properties, stations in the Antarctic below 50 degrees south were excluded [35–37]. The differences are often explained by the adaptation or acclimation of polar phytoplankton to extreme environmental characteristics or because of alterations in the relative abundances and characteristics of other optically-significant constituents resulting from particular geographical settings, specifically in the Southern Ocean [35,37–45]. In order to promote the greater variability of the Chla within the sunlit surface layer, a 9-point logarithmic depth grid was defined between the surface and 300 m to represent the greater near-surface variability: 5 m, 8.34 m, 13.92 m, 23.23 m, 38.75 m, 64.63 m, 107.81 m, 179.84 m and 300 m. For each station, multiple measurements occurring in a same depth point were averaged. From the initial longitude, latitude and date of the HPLC measures, 6807 stations were found and then reduced to 3903 stations which are collocated with satellite observations whose resolution is 4 km × 4 km. The stations that contained more than 50% missing pigment values were excluded, resulting in a final total of 1614 retained stations. The geographical distribution of the stations is shown in red in Figure 1.

A separated database has been used in the last section of the paper to test the proposed methodology. The Tara Oceans HPLC pigment concentration database from the Tara Oceans Expedition [26] contains HPLC measurements for several pigments at different depths, from which we select the data corresponding to the 6 pigments we are interested in (Chla, fucox, perid, 19hex and zeax). The measurements are composed of 211 stations distributed over the globe, which were combined into 143 stations according to the satellite resolution and excluding Antarctic stations. This dataset has been cleaned in the same way as MAREDAT, resulting in 66 stations whose geographic distribution is shown in green in Figure 1.

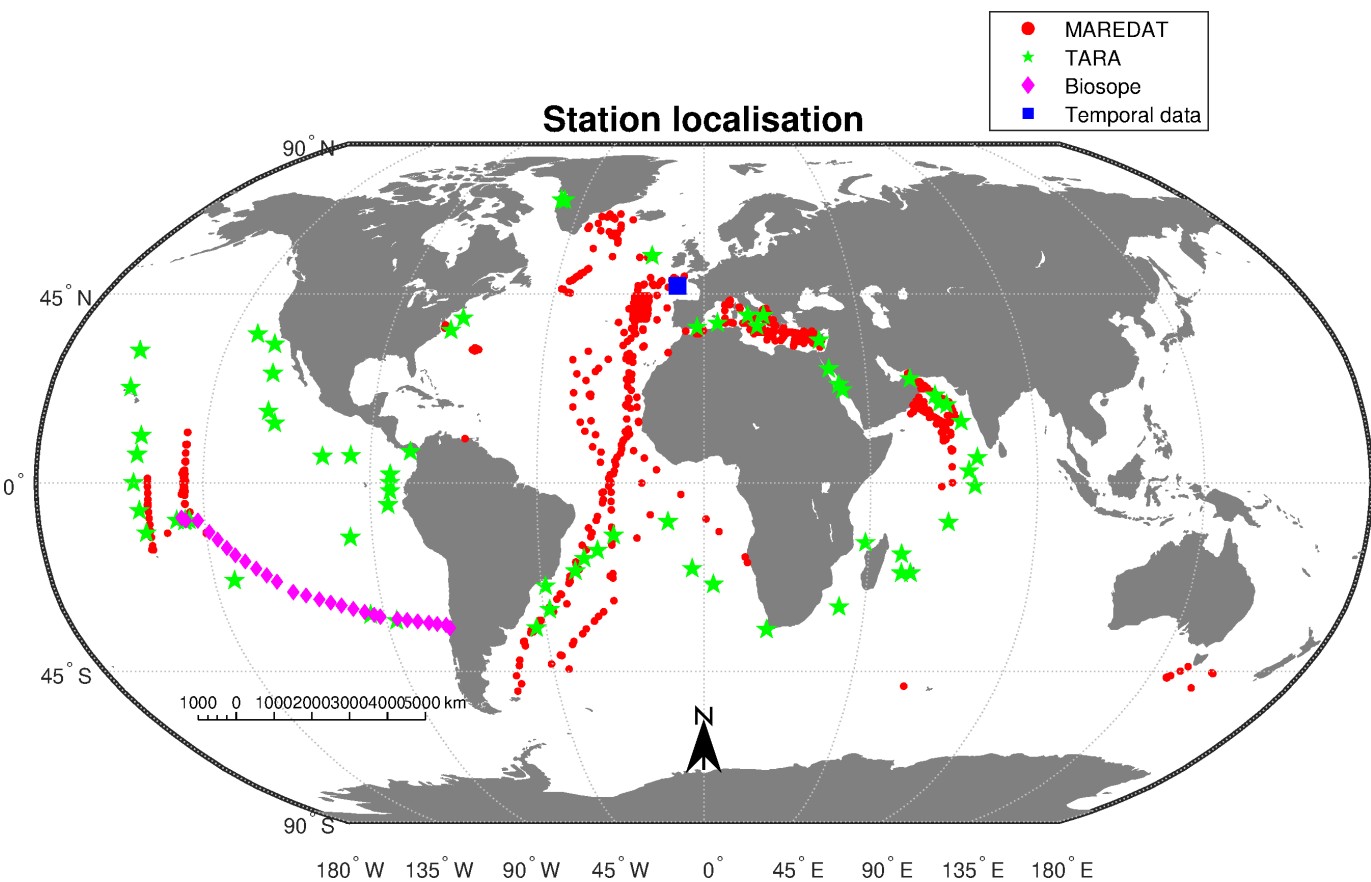

**Figure 1.** Geographical repartition of the stations. Red dots represent the repartition of the 1614 stations from MAREDAT constituting the training set, green stars represent the repartition of the 66 stations from Tara constituting the test set. The magenta diamonds represent the Biosope trajectory, a subset of the MAREDAT dataset, and the blue square indicates the location where satellite data were obtained in order to test the developed method.

### 2.1.2. Satellite Observations

The ocean colour satellite data originates from the Globcolour project, carried out by the European Space Agency (ESA), consists of creating and maintaining a long time-series of ocean color data from satellite measurements (from 1997 till present). This database is the result of the fusion of data from various satellite sensors: Sea-viewing Wide Field-of-view Sensor (SeaWiFS), Moderate Resolution Imaging Spectroradiometer (MODIS), Visible Infrared Imaging Radiometer Suite (VIIRS), Medium Resolution Imaging Spectrometer (MERIS), and Ocean and Land Colour Instrument (OLCI).

The sensors measure the backscatter and spectral absorption coefficients of light by the ocean, and the reflectance is then calculated from these parameters. The reflectances are generated by each sensor from level 2 data (data pre-processed according to sensor and geophysical parameters). The reflectances are then merged by taking a weighted average of each sensor output. Meanwhile, Sea Surface Temperature (sst) was obtained from the Advanced Very High-Resolution Radiometer (AVHRR) instruments on board of the National Oceanic and Atmospheric Administration (NOAA) 5.3 [46,47]. The satellite data have undergone quality and flag check and are generated with a spatial resolution of 4 km and a temporal resolution of a day.

Eleven satellite measurements were proposed to be used for retrieving the 6 pigment concentration profiles that constitute the pigment database: Remote sensing reflectances at 4 wavelengths (RRS412, RRS443, RRS490, and RRS555), satellite Chla (chla_sat), Sea Surface Temperature (SST), light attenuation coefficient at 490 nm (KD490), depth of the euphotic layer (ZEU), depth of the warmed layer (ZHL), photosynthesis available radiation (PAR) and its coefficient of attenuation (KDPAR).

The choice of the satellite variables is based on the findings of previous studies [13,14]. It has been shown in these studies that surface Chla and the euphotic depth (ZEU) are the main variables explaining the vertical variability of the Chla in the water column. However, since we are dealing with several pigments, it is primordial to use the surface reflectance at different wavelengths rather than only satellite-derived Chla to consider the influence of other pigments' variability on the satellite-detected signal. Physical factors are also investigated to take into account the influence of light (PAR, KDPAR, KD490, ZEU) and heating (SST, ZHL) on this vertical variability. In order to validate our use of the Satellite data, we compared the Chla in-situ data (Section 2.1.1) to the Globcolor Chla product. The calculated regression coefficient and the Spearman correlation were 0.67 and 0.77, respectively.

The two separate datasets were merged into a final reduced database colocating the in situ observations with the satellite data. Finally, the database subsequently used for the construction of the method, noted $D$, of dimension $(1614, 65)$, where 1614 is the number of in situ profiles (stations) colocalised with satellite images, noted $z_i$, and 65 the number of variables, consisting of 54 in-situ HPLC pigment variables (6 pigments, 9 depths each) and 11 satellite variables.

### 2.1.3. Combined Dataset

The dataset resulting from the merging of the two databases is of high dimension, due to the inclusion of the concentrations of the six pigments at nine depths, and show scattered data as it can be seen in Table 1. The omission of localization elements such as the latitude and longitude in this study is tied to a lack of sufficient data to prevent over-fitting. Furthermore, since phytoplankton are associated with nonlinear population dynamics [48], there exist strong nonlinear relationships among the different concentrations of photosynthetic pigments. We are therefore working on high-dimensional and scattered pigment data, with strong nonlinear relationships. The development of a method for in-depth reconstruction then requires the choice of a suitable technique that can manage these nonlinear relations.

**Table 1.** Missing data for each pigment (among the 9 depths) and for the satellite variable of the experimental dataset $D$.

| Pigment | Missing Data (%) |
|---------|------------------|
| Chla | 30 |
| DVChla | 48 |
| 19hex | 32 |
| fucox | 30 |
| perid | 32 |
| zeax | 40 |
| Satellite data | 70 |

### 2.2. Inverse Method: From Satellite Data to Vertical Profiles

In order to infer the vertical distribution from vertical profiles, we need to enchain different methodological phases that rely on Artificial Neural Networks and dimension reduction techniques. These methods are briefly outlined in this section, before detailing the specific implementation.

### 2.2.1. Algorithms

Neural approaches can be used to study nonlinear interactions within complex self-adaptive systems, such as marine ecosystems in relation with remote sensing measurements. Unsupervised approaches make it possible to extract these nonlinear relationships without any *a priori* assumptions.

The Self-Organizing Maps (SOM) [49] are unsupervised neural networks, whose objective is to cluster a high dimensional dataset $D \in R^n$ into a discrete representation in reduced dimensions, generally on a two-dimensional neural grid called a "map". This grid layout allows the introduction of the notion of neurons' neighborhood during the clustering so that two clusters that are near on the topological map gather similar data, thanks to the topological ordering of the map. They have the advantage of having high interpretability and make it possible to find relationships between the distribution of data on the map and the main explanatory variables. This is particularly useful in the case of complex and noisy data—as it is the case with climatology/oceanography data where they have been used in a large variety of studies [50,51].

After training, each cluster is defined by a referent vector $W_C$, which represents the mean value of the data assigned to it, and by its position on the topological map, which indicated the clusters which are close to it. The attribution of a data $Z$ to a class is made by comparing it to the set of referent vectors $\{W_C; C \in SOM\}$ and attributing them to the nearest referent vector $W_c$ according to the Euclidean distance ($C$ is called *Best-Matching Unit* or BMU) (1):

$$BMU(Z, SOM) = argmin_{C \in SOM} \sqrt{\sum_{i=1}^{n} (Z_i - W_{C_i})^2}, \tag{1}$$

where $Z \in R^n$. The SOM can be used in the context of completing missing data [52] by considering a modification of this distance. In that case, the projected vectors $Z$ can have components $Z_i$ whose values are missing. Under these conditions, the distance between a vector $Z$ and the referent vectors $W_c$ of the map is the Euclidean distance that considers only the existing components (the Truncated Distance or TD hereinafter). The use of the TD allows for taking into account the information embedded in the incomplete data.

The Iterative Completion SOM (ITCOMPSOM) method is an iterative data completion method derived from the SOM. When a data vector presents missing values, the method uses a modified TD, denoted $TD_s$ as seen in Equation (2). The modified TD makes use of the correlations between the missing variables and those present to weight the Euclidean distance so that the variables most correlated to the missing values will more strongly influence the attribution to a cluster:

$$TD_s^c(Z, W_c) = \sum_{i \in avail.} \left( \left( 1 + \sum_{j \in missing} (cor_{ij})^2 \right) \times \left( Z_i - W_{C_i} \right)^2 \right),$$

where *avail.* corresponds to the components of $Z$ without missing values, while *missing* to those with missing values. The correlations $cor_{ij}$ are calculated pairwise between all variables over the training data set before applying the method.

Furthermore, ITCOMPSOM iteratively completes the dataset, imputing the missing values of a data vector several times during the iterations, by training successively bigger topological maps, which combine previously completed data and new data with missing values at each iteration. This method allows a better data completion than the basic SOM method, for data with up to 75% missing data. Moreover, it is adapted to the completion of oceanographic data in which the variables are linked [23,53].

Finally, we also used Principal Component Analysis (PCA) [54], which is an orthogonal linear transformation of a dataset that projects the values onto new axes that best fit the data. These new axes are selected to explain a maximum amount of variability of the initial data. It can also be seen as a filtering tool, the first axes representing most of the information embedded in the data set, the remaining axes being associated with dataset noise. The specific number of modes was selected by cross-validation and are presented in Section 3.1.

2.2.2. Sat2profile Methodology

The aim of Sat2profile is to retrieve the vertical profile using the satellite data only. Due to the huge number of missing data and the level of noise occurring in the observation data, this requires a complete methodology taking each problem into account. *Sat2Profile* can be split into three main phases:

1.  Selecting an initial set of explanatory variables proposed by an expert.
2.  Completing the missing data occurring on the pigment observations using ITCOMP-SOM.
3.  Applying a PCA to filter and compress the vertical profiles to be retrieved by *Sat2Profile*. During this phase, two hyper parameters are determined: the number of PCA ($n_{axes_i}$) and the size of the map.

At the end of these 3 phases, we perform a variable selection. We fix the hyper parameters $n_{axes_i}$ and the size of the map, and we test all the possible combinations of explanatory variables reiterating the *Sat2Profile* inversion for each subset. Figure 2 summarizes the methodological process.

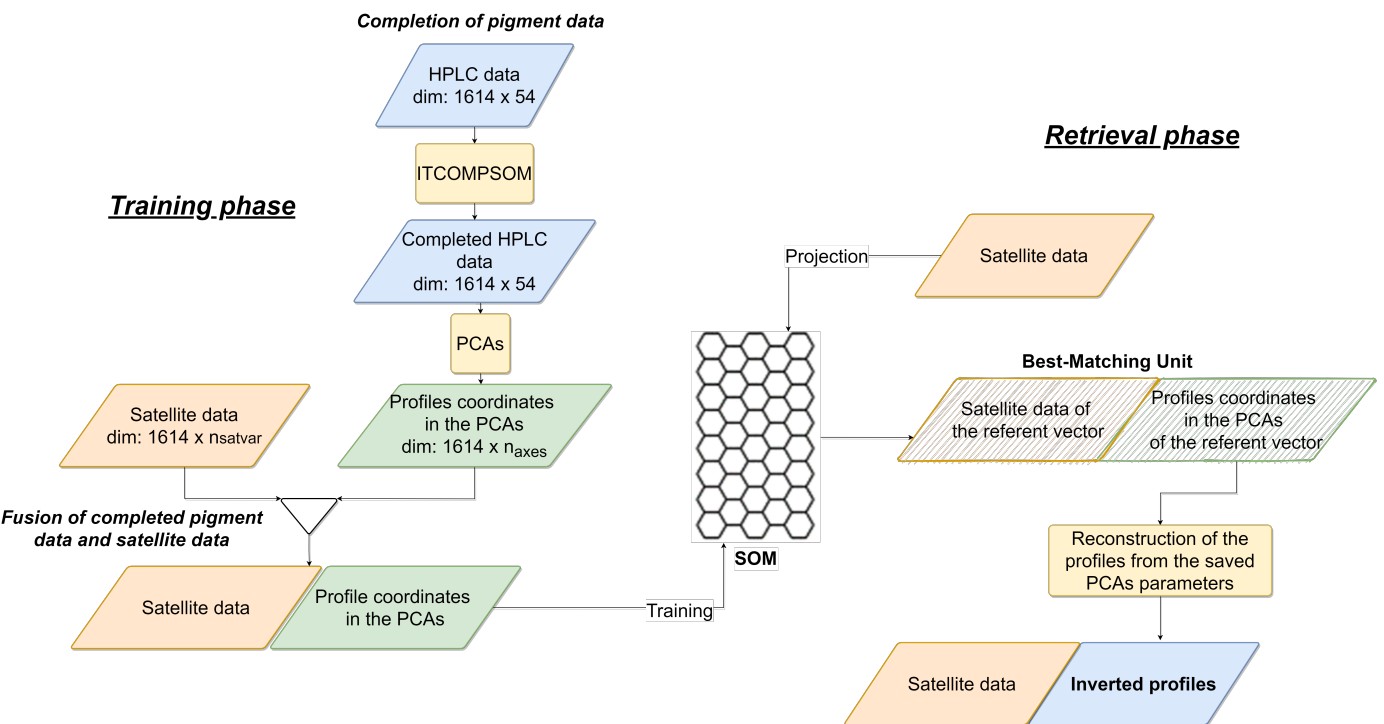

**Figure 2.** Flow diagram of the *Sat2Profile*. A 500-fold cross-validation was effectuated on the training data.

In our study, the different phases were implemented in the way presented below.

*1st Phase*: we chose to use the satellite variables RRS412, RRS443, RRS555, KD490, ZEU, and ZHL that we expected to have the best ability to retrieve the vertical distributions of the pigment concentrations. As described in Section 2.1.2, the surface reflectance at different wavelength is used to consider the influence of the pigments' variability on the satellite-detected signal. KD490, ZEU and ZHL are also used to take into account the the sun light and heating effects.

*2nd Phase*: The learning dataset $D$ has two distinct components: the satellite data that can have missing data and the pigment profiles. The pigment profiles were completed using ITCOMPSOM. The most complete part of the dataset (106 observations from the 1640, across the globe) is set aside as a validation set. Parts of these data were artificially masked. The ITCOMPSOM method was trained with the rest of the dataset and used to complete the validation set. The completed data and the corresponding observed data were compared

computing $R^2$ and RMSE. This process was repeated a large number of times (500 times) and an average assessment of completion was obtained, shown in Table 2.

*3rd Phase*: The completed pigment data were collocated with the satellite measurements and combined into a single dataset. Then, a smoothed version of the pigment dataset was constituted by using PCAs. For a given number of axes $n_{axes}$, a learning dataset was constituted with 1614 lines and $11 + n_{axes}$ columns corresponding, respectively, to the satellite and the smoothed PCA profiles. All the variables of the resulting dataset are centered-reduced and are used as a training set for a SOM. The cross validation (described in Section 2.2.3) resulting from the 9 experiences (dimension of the profiles) allows the determination of $n_{axes_i}$.

Finally, after having selected the optimal number of axes to keep, we analyzed the whole *Sat2Profile* methodology, testing all the combinations of the 11 satellite variables to be used as inputs allowing the best retrieval of pigments' vertical profiles. The exact hyperparameter values are provided in the code (https://github.com/AgathePuissant/SOM_PCA (accessed on 1 March 2020)). At that time, we found that the 6 selected variables (RRS412, RRS443, RRS555, KD490, ZEU and ZHL) were the optimal combination of variables to be used.

**Table 2.** Validation results for the completion of the data by ITCOMPSOM.

| Pigment | $R^2$ | RMSE (mg m$^{-3}$) |
|:---:|:---:|:---:|
| Chl-A | 0.70 | 0.181 |
| DVChl-A | 0.78 | 0.016 |
| 19-Hex | 0.64 | 0.032 |
| Fucox | 0.74 | 0.035 |
| Perid | 0.53 | 0.005 |
| Zeax | 0.73 | 0.014 |

2.2.3. Methodological Workflow

Training Phase

First, the training dataset was completed using the ITCOMPSOM method. A PCA was performed on the matrix of in situ data for each pigment of dimension (1614,9). These PCAs resulted in 9 principal components for each pigment. A certain number $n_{axes}$ of these principal components were kept (the precise number for each pigment was chosen through optimization), resulting in 6 pigment datasets of dimensions $(1614, n_{axes_i})$, with $i \in [1 \ldots 6]$. The pigment data were colocated with the satellite measurements and combined in a single dataset. All the variables of the resulting dataset were centered-reduced and were used as a training set for a SOM.

Retrieval Phase

After the initial training, the SOM can be used to reconstruct the missing $\sum_i n_{axes_i}$ variables of in situ-data from the available $n_{satvar}$ variables of satellite-derived data. Each observation was assigned to its Best Matching Unit, the neuron in the map whose referent vector was the closest in the Euclidean sense (1). The missing data were then replaced by the values of the corresponding components of the assigned referent vector. The PCA coordinates of the profiles were retrieved from the satellite data input, and then the profiles were reconstructed in the data space using the determined PCA parameters.

Cross-Validation of the Model

To assess the performance of the method, a 500-fold cross-validation procedure has been set up: the preprocessed database used is randomly segmented into 500 blocks. In each iteration, 499 out of the 500 blocks are used as a validation set. The pigment data from the validation set is masked, only the satellite variables data are kept and used to infer the missing values.

The SOM is trained on the training set, and the retrieval procedure is applied to the validation set. The estimated pigment data from the validation set is compared to the corresponding observed data that had been masked beforehand. This process is repeated on the 500 blocks.

The performance of the retrieval is assessed by computing the $R^2$ (2), Root-Mean Squared Error ($RMSE$) (3) and Spearman correlation coefficient (4) between each observed and estimated profile. They are then averaged for each pigment.

$$R^2(Obs_i, Est_i) = 1 - \frac{\sum_{j=1}^{n}(Obs_{ij} - Est_{ij})^2}{\sum_{i=1}^{n}(Obs_{ij} - \overline{Obs_i})^2} \, , \ i \in [1, m] \tag{2}$$

$$RMSE(Obs_i, Est_i) = \sqrt{\frac{\sum_{j=1}^{n}(Obs_{ij} - Est_{ij})^2}{n}} \, , \ i \in [1, m] \tag{3}$$

$$\rho_{Spear}(Obs_i, Est_i) = 1 - \frac{6\sum_{j=1}^{n}d^2}{n(n^2 - 1)} \, , \ i \in [1, m] \tag{4}$$

where $d$ is the rank difference among the vectors, $n$ the number of components in the vector ($n = 9$ because the profiles are composed of 9 depths) and $m$ the number of observations in $D$ ($m = 1614$).

The $R^2$ and $RMSE$ are computed from the linear regression between the observed and estimated values for each profile and allows the quantification of the error committed during the profile retrieval. The Spearman correlation coefficient accounts for nonlinear relationships among variables, and thus allows an assessment of the correspondence of the shapes of the estimated versus observed profile.

### 2.2.4. Test of Spatial and Temporal Coherence

Once the inversion method has been implemented, the results obtained must be spatially and temporally consistent. To test the results of the method on spatially varying data, the inversion method was applied to observations in a particular ocean cruise transect. The Biosope cruise transect (http://www.obs-vlfr.fr/proof/vt/op/ec/biosope/bio.htm (accessed on 1 March 2020)) was selected based on the quantity of satellite data available to invert pigment profiles from. The Biosope transect is composed of 49 stations, 28 of which contain enough satellite data to perform an inversion. This transect data come from the training set and therefore was used to verify the spatial consistency of the results from our inversion method. On the other hand, to validate the consistency over time of the data obtained by inversion, we selected a station located in a temperate zone (47°N, 8°W) and therefore where phytoplankton show a well-marked seasonality. The weekly satellite data (averaged over 8 days) observed during the year 2019 from January to December were extracted from a $6 \times 6$ pixel box around the location coordinates. Pigment profiles were inverted from satellite data and then the profiles were spatially averaged for each week, resulting in 46 weekly average pigment profiles.

## 3. Results

### 3.1. Parameters of the Method

The data were completed using the ITCOMPSOM method with a two-dimensional hexagonal grid with a final size of $27 \times 15$ (405 neurons) on the SOM and 10 iterations. The SOM consists of the same structure of a two-dimensional hexagonal grid with a size of $27 \times 15$ (405 neurons), determined heuristically by taking into account the number of observations in the training set and the number of observations per class, to have a good distribution of data on the neural map. Cross-validation experiments of the performance of the method helped to determine the number of PCA coordinates to keep for each pigment. The first two PCA coordinates were kept for each pigment, corresponding to between 69% and 82% of the explained variance depending on the pigment. After cross-validating the

method for every combination of the considered eleven satellite variables, the six selected variables were RRS412, RRS443, RRS555, KD490, ZEU and ZHL.

### 3.2. Cross Validation Performance

The results of the cross-validation of the method using the PCA preprocessing with two axes were compared with the results of the cross-validation of the method without the smoothing of the profiles by the PCAs given in Table 3. The average $R^2$ and average Spearman's correlation coefficient per profile increase with the use of profile smoothing by PCA, while the average RMSE per profile decreases. As an example, for fucoxanthin, the average $R^2$ per profile increases from 0.4 to 0.83 with the use of PCA smoothing in the inversion method. On average, the Spearman's per profile correlation coefficient increased by 0.26, the $R^2$ per profile increased by 0.31, and the RMSE per profile was divided by 2.17. Globally, for the method using a PCA reduction, the average $R^2$ per profile ranges from 0.68 to 0.83, and the average Spearman correlation coefficients per profile range from 0.77 to 0.84.

**Table 3.** Cross-validation results for the method without PCA preprocessing, and with PCA preprocessing (two axes).

| | Mean Spearman Correlation | | Mean $R^2$ | | Mean RMSE (mg m$^{-3}$) | | Mean RMSE (% of Mean Concentration) | | Mean Concentration (mg m$^{-3}$) |
|---|---|---|---|---|---|---|---|---|---|
| | Without PCA | With PCA | Without PCA | With PCA | Without PCA | With PCA | Without PCA | With PCA | |
| **Chla** | 0.65 | 0.81 | 0.56 | 0.81 | 0.083 | 0.036 | 36.4 | 15.8 | 0.2280 |
| **DVChla** | 0.475 | 0.79 | 0.42 | 0.68 | 0.011 | 0.006 | 43.5 | 23.7 | 0.0253 |
| **19hex** | 0.62 | 0.82 | 0.53 | 0.81 | 0.02 | 0.008 | 35.4 | 14.2 | 0.0565 |
| **fucox** | 0.52 | 0.84 | 0.4 | 0.83 | 0.012 | 0.005 | 40.3 | 16.8 | 0.0298 |
| **perid** | 0.42 | 0.78 | 0.34 | 0.76 | 0.002 | 0.001 | 45.5 | 22.7 | 0.0044 |
| **zeax** | 0.59 | 0.77 | 0.57 | 0.81 | 0.01 | 0.005 | 30.2 | 15.1 | 0.0331 |

To assess the order of magnitude of the information lost by the PCA smoothing, the initial profiles have been compared before and after the PCA preprocessing with two axes, using the RMSE averaged over all the observations for each pigment. The results are presented below in Table 4 along with the RMSE estimates from the cross validation, and represent the uncertainties associated with each estimated pigment vertical profile. Clearly, the percentage of errors for the two steps, PCA and SOM, have the same order of magnitude.

**Table 4.** Mean RMSE results for the PCA step of the method and the SOM step of the method.

| | PCA | | SOM | |
|---|---|---|---|---|
| | Mean RMSE (mg m$^{-3}$) | Mean RMSE (% of the Mean Concentration) | Mean RMSE (mg m$^{-3}$) | Mean RMSE (% of the mean Concentration) |
| **Chla** | 0.046 | 20.2 | 0.036 | 15.8 |
| **DVChla** | 0.006 | 23.7 | 0.006 | 23.7 |
| **19hex** | 0.011 | 19.5 | 0.008 | 14.2 |
| **Fucox** | 0.005 | 16.8 | 0.005 | 16.8 |
| **Perid** | 0.001 | 22.7 | 0.001 | 22.7 |
| **Zeax** | 0.005 | 15.1 | 0.005 | 15.1 |

### 3.3. Test Performance

Once the method has been trained on the ITCOMPSOM completed and PCA preprocessed data, the retrieval procedure was applied to satellite data colocated with the 66 Tara dataset stations. The Tara profiles were completed using ITCOMPSOM to allow the comparison between observed and estimated profiles. The estimated pigment profiles were compared to the completed observed ones. The results are shown in Table 5. The comparison criteria are in the same order of magnitude as the results of the cross-

validation experiment. These results suggest a good generalization capability of the method to exterior data.

**Table 5.** Results of the inversion of the Tara test set using the method with PCA preprocessing (two axes).

| | Mean Spearman Coefficient | Mean $R^2$ | Mean RMSE (mg m$^{-3}$) | Mean RMSE (% of Mean Concentration) |
|---|---|---|---|---|
| Chla | 0.75 | 0.74 | 0.042 | 18.4 |
| DVChla | 0.74 | 0.65 | 0.012 | 47.4 |
| 19hex | 0.78 | 0.74 | 0.008 | 14.2 |
| fucox | 0.82 | 0.79 | 0.003 | 10.1 |
| perid | 0.72 | 0.72 | 0.001 | 22.7 |
| zeax | 0.80 | 0.86 | 0.007 | 21.1 |

*3.4. Spatial and Temporal Coherence*

The pigment profiles of the Biosope cruise trajectory were estimated from the daily satellite data using our method. The results for the main pigment (Chla) and a secondary pigment (DVChla) are shown in Figures 3 and 4. In these figures, as the cruise trajectory crosses the Pacific Ocean longitudinally, we chose to represent the pigment concentration values along the longitude on the *x*-axis and the depth values on the *y*-axis. The profiles, smoothed using PCAs, which are represented in Figures 3a and 4a, are the final profiles that we aimed at retrieving from satellite data. The inverted profiles are represented in Figures 3b and 4b, the black areas corresponding to the longitudes where there were no matched satellite data available for any of the six selected satellite variables. Figures 3c and 4c show the difference between observed and estimated profiles.

In Figures 3b and 4b, we show the profiles estimated by inversion, which can be compared with Figures 3a and 4a. Globally, we find the same zones and the same depths for the concentration maxima. The same pattern of the maximum concentration depth as a function of longitude is found both in the estimated and observed profiles, i.e., close to the surface in the west, then reaching deep depths between 107.81 m and 179.85 m at intermediate longitudes and again close to the surface in the eastern longitudes. However, some profiles are overestimated, other underestimated, which are respectively shown in red and blue in Figures 3c and 4c. This test of the inversion method on the Biosope cruise trajectory satisfactorily accounts for the inter-pigment dynamics along a continuous spatial observation. The spatial coherence of the trajectory is preserved after the inversion from satellite data.

The weekly pigment profiles in the ocean area (47°N, 8°W) were inverted from satellite data by our method for the year 2019. The inversion was performed using satellite data not included in the training dataset. Only satellite data were available at this location, but the temporal characteristics of phytoplankton are known: the region corresponds to the North Atlantic Biogeochemical province, with a temperate climate and a seasonal variation of phytoplankton. Therefore, a spring bloom of phytoplankton is expected. This inversion thus allows us to test the method on new data and to verify the temporal coherence of the results obtained with the environmental characteristics. We show the results for the estimated Chla, fucox, and zeax profiles with respect to time. The Chla concentration represents the occurrence of the phytoplankton as a whole, and the fucox and zeax represent the composition of the phytoplankton community. These two secondary pigments are indicators of two main groups of phytoplankton, fucox being a diagnostic pigment for the diatoms [30] and zeax being a diagnostic pigment for the prokaryotes [33,34].

Figure 5 shows Chla profiles as a function of depth and time (in weeks). Between weeks 10 and 18, which corresponds to mid-March to early May, the Chla reaches high concentrations in the water column with a maximum at the surface between 5 and 8 m. Following that, the surface Chla concentration decreases, showing a DCM between 23 and 64 m. As seen in Figure 6, there is a concentration peak of fucox at a depth of about forty meter at the same time as the Chla peak, between weeks 10 and 18. In Figure 7, we observe a different

dynamic for the zeax concentration with respect to the two other pigments: the concentration peak occurs later during weeks 19–37 corresponding to the late spring/summer seasons. The increase of zeax happens at the surface layers (between 5 and 23 m).

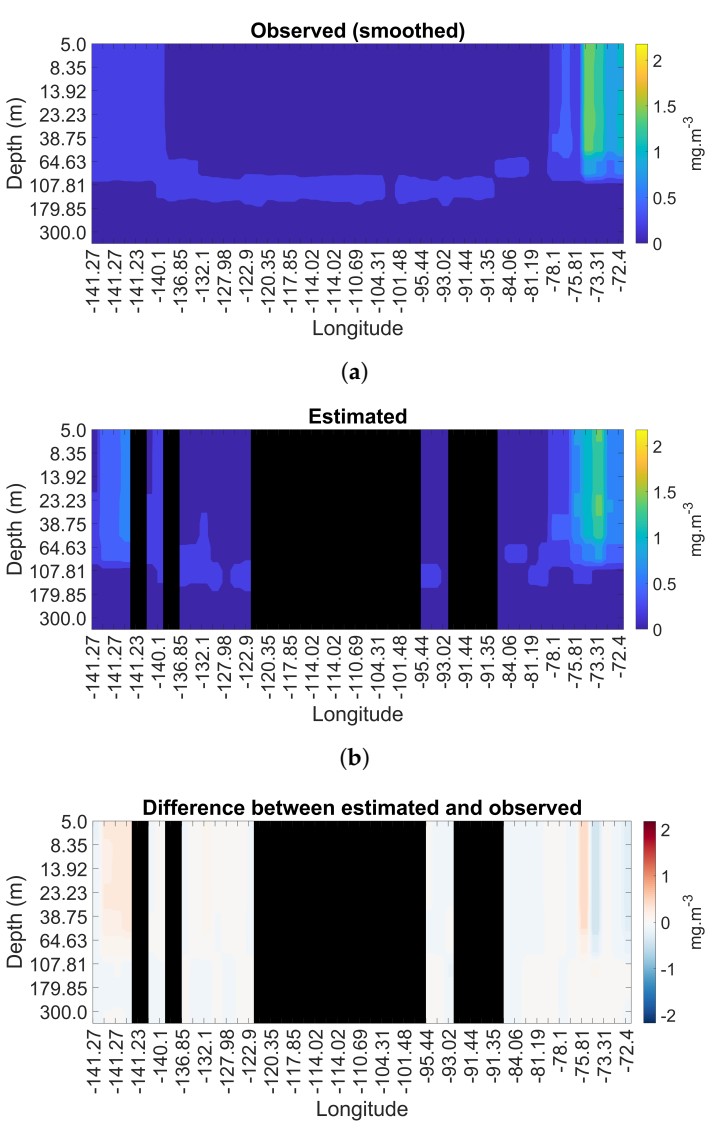

**Figure 3.** Result of the inversion of Chla profiles from the satellite data of the Biosope trajectory. (**a**) Smoothed observed Chla profiles; (**b**) estimated Chla profiles; (**c**) difference between estimated and observed.

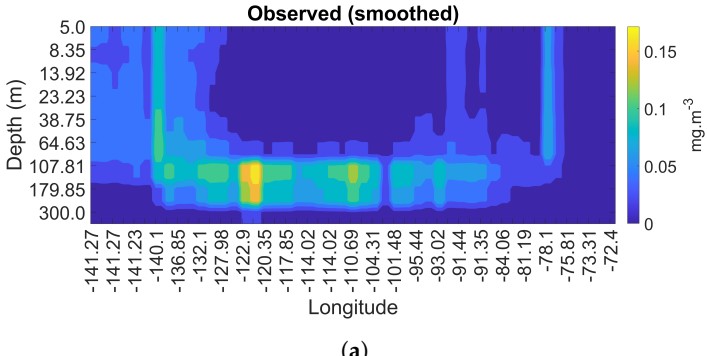

**Figure 4.** *Cont*.

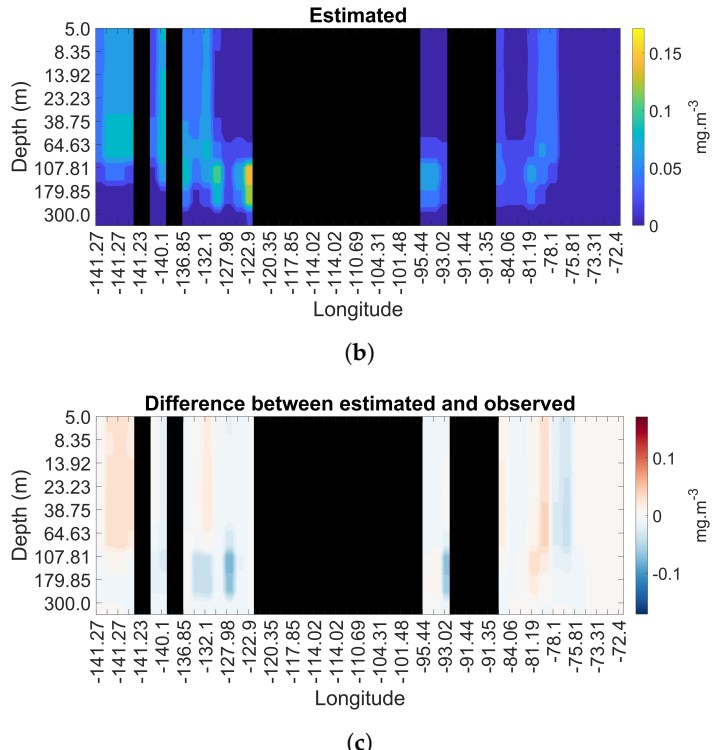

(b)

(c)

**Figure 4.** Result of the inversion of the DVChla profiles from the satellite data of the Biosope trajectory. (**a**) Smoothed observed DVChla profiles; (**b**) estimated DVChla profiles; (**c**) difference between estimated and observed.

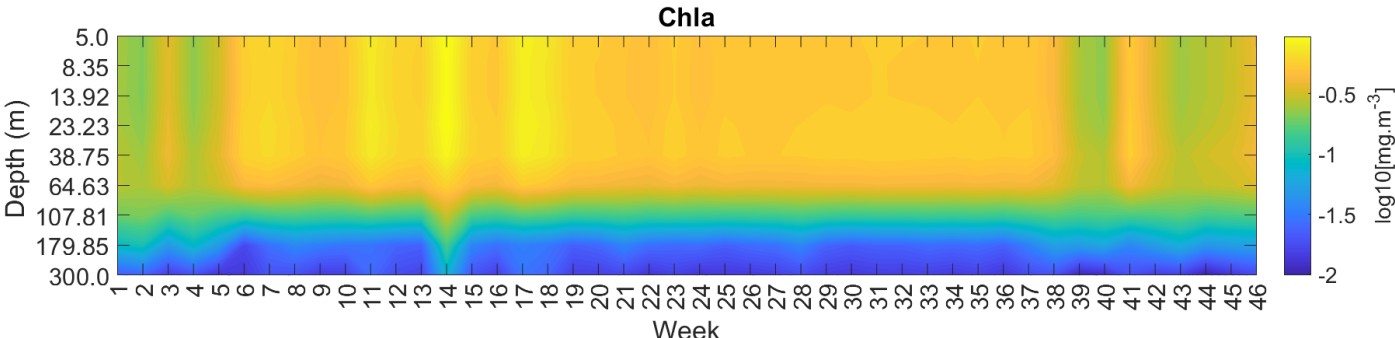

**Figure 5.** Chla inverted profiles over time.

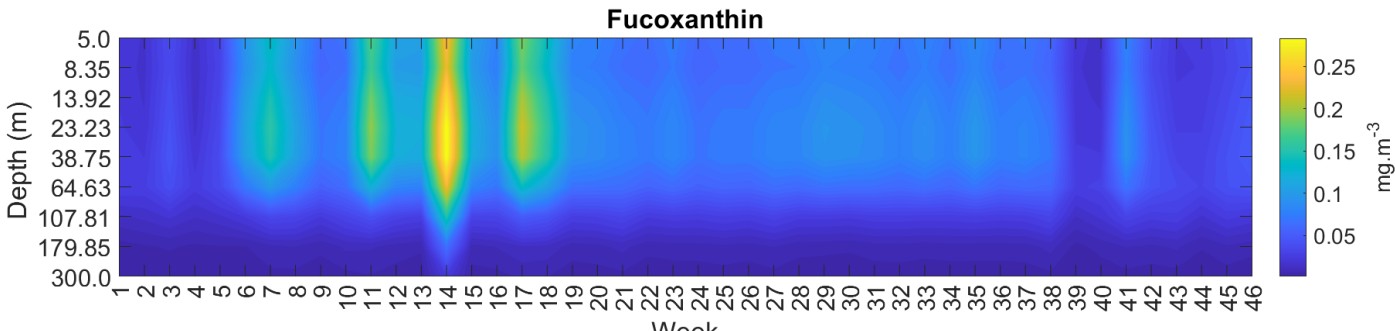

**Figure 6.** Fucox inverted profiles over time.

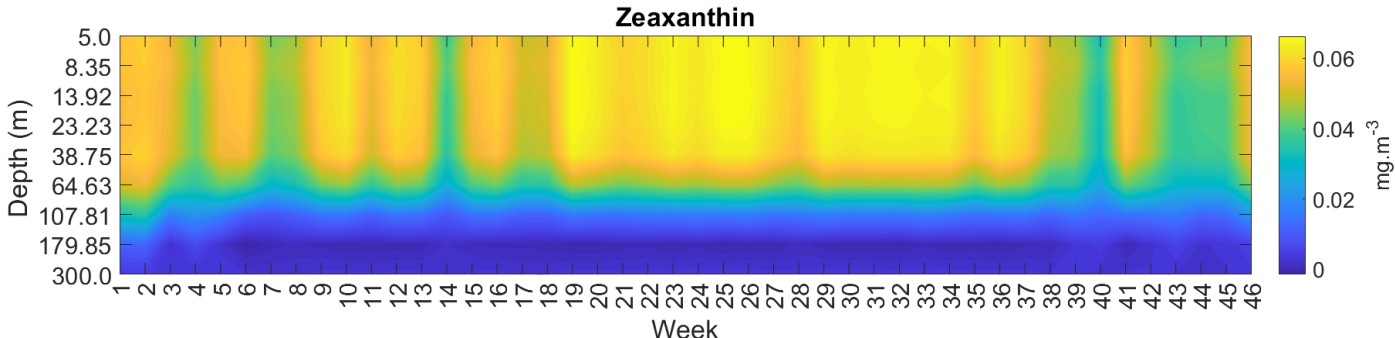

**Figure 7.** zeax inverted profiles over time.

## 4. Discussion and Conclusions

We presented, in this paper, robust estimations of the vertical variability of six phytoplankton pigments (Chla, fucox, 19hex, perid, zeax and DVChla) from the surface to a depth of 300 m, using satellite surface measurements at high spatial (global, 4 km) and temporal (daily) resolution. These estimations are derived from a new machine learning methodology proposed in this paper, *Sat2Profile*, based on a SOM, and trained and validated using the fusion of an in situ global HPLC database, MAREDAT, and an ocean colour satellite database. After a series of cross-validations and checking the coherence of the results, a validation experiment was performed on a new database introduced as a test set from Tara Oceans measurements. The different experiments show a satisfying performance. The different regression coefficients $R^2$ between observed and estimated vertical profiles of pigment concentration and the Spearman correlation coefficient are greater than 0.7. The reconstruction of the 3D distribution of phytoplankton pigments is an innovative result that gives a better understanding of the PFTs distribution in the water column.

Works attempting to predict vertical pigment profiles from surface data targeted the Chla and were based on the surface Chla and/or assigned with other physical factors such as SST and currents ([13,14,17–20]). However, during the optimization process of *Sat2Profile*, we showed that the problem is more complex when dealing with different pigments at the same time, each with their own particular variability. SST and Chla surface information were not enough to estimate the vertical profile of the pigments. Therefore, several bio-optical parameters, such as remote sensing reflectance at several wavelengths, and the information about the euphotic layer were essential to infer pigment vertical variability from surface data. The necessity of having euphotic depth as an input aligns our study with the reasoning of [13]. In addition, in [55], the authors proved that optical and radiometric information are effective indicators of the vertical dynamics of pigments. Estimating phytoplankton pigment variability using a temporal dataset of satellite data within the North Atlantic biogeochemical province showed that pigments such as Zeaxanthin and Fucoxanthin exhibit different temporal variability over time. Furthermore, the depth of the pigment concentration maximum is not the same for each pigment; this was observed in in-situ studies [56] and have been also observed in the MAREDAT database. These findings can be related to the community shift in response to seasonal changes and variations of environmental factors. The fucox peak concentrations indicate a bloom dominated by diatoms. The overall low zeax concentration highlights that the fraction of prokaryotes at this time is limited. Later, with the heating of the surface layer at the beginning of the summer until the end of September, the fucox decreases while the zeax remains the same. In such events of stratification of the water column in response to higher SST, prokaryotes are the most favored by these environments [57,58]. This analysis of pigments dynamics along time is consistent with studies done in the North Atlantic Biogeochemical region [59,60].

The Biosope experiment to reconstruct the pigment variability along the ship transect using *Sat2Profile* showed satisfying concordance. The transect crosses a region characterized by the presence of the southern sub-tropical gyre, which is known by its ultra-oligotrophic

environment. In other terms, this nutrient poor environment is represented by the lowering of the overall Chla concentration in this gyre and deepening of the DCM as seen in the in-situ database. *Sat2Profile* estimation of Chla and DVChla shows an interesting ability of the method to capture the deep DCM and the variability of pigments using surface satellite data in that region of the southern Pacific.

Indeed, the inter-pigment relationships are specific to regions and to trophic states ([13]), and the variability of these pigments is capable to reflect the influence of environmental factors such as nutrient dependency and water masses on the phytoplankton community structure ([61,62]).

Uitz et al. and Sauzede et al. [14,18] exploited the data obtained by HPLC to determine the different phytoplankton size classes occurring in the water column based on their contribution to the total Chla [14]. The pigment variability seen in our previously described analysis can be compared to the results of both studies. Indeed, fucox is usually used to estimate microphytoplankton relative abundance and zeax for picophytoplankton. The variability of these two size classes is seen to be antagonistic in the work of [14,18]; more microphytoplankton in a Chla-rich water column, and more picophytoplankton in poor oligotrophic waters. This corresponds also to the variability of fucox and zeax in our temporal study.

However, the difference brought by the presented method is that PSC estimations in [14,18] were constrained by the empirical relationships between Chla and secondary pigments and by a priori hypotheses on the shape of the vertical pigments profiles [15]. In order to avoid biases introduced with these inter-pigment empirical relationships, *Sat2Profile* aims to estimate phytoplankton pigments as a first step. In a later stage, *Sat2Profile* unfolds the opportunity to observe phytoplankton groups derived from pigments and to assess the retrieval of these PFTs from empirical relationships.

The method we present is globally applicable (excluding the Southern Ocean) and generates daily products from 1997–present; this opens the way for multiple new studies. However, several limitations cannot be denied. There are uncertainties resulting from the error propagation in the *Sat2Profile*: through the data completion and the loss of information during the PCA filtering until the retrieval from satellite data. These errors were quantified and addressed in this paper. However, the information retrieved using *Sat2Profile* is one step toward closing the gap of knowledge in the distribution of phytoplankton groups, especially below the surface where sampling of phytoplankton diversity measures has been very scarce.

The existence of direct links between pigment concentrations and phytoplankton functional types implies that we can use this approach to attempt to study their global vertical distribution. This would improve the global spatio-temporal monitoring of the biological pump, crucial in constraining our estimations of the ocean's absorption capacity in a changing climate.

**Author Contributions:** Conceptualization, S.T., R.E.H. and A.A.C.; methodology, S.T., R.E.H., A.P. and A.A.C.; software, A.P.; validation, A.P.; formal analysis, S.T., R.E.H., A.P. and A.A.C.; investigation, S.T., R.E.H., A.P. and A.A.C.; resources, S.T. and C.B.; data curation, R.E.H. and C.B.; writing—original draft preparation, A.P.; writing—review and editing, S.T., R.E.H., A.P. and A.A.C.; visualization, A.P.; supervision, S.T., R.E.H., C.B. and A.A.C.; project administration, S.T. and R.E.H.; funding acquisition, S.T. All authors have read and agreed to the published version of the manuscript.

**Funding:** This project was carried out with the support of the Sorbonne Center for Artificial Intelligence (SCAI) of Sorbonne University. A.P. is supported by l'Ecole Universitaire de Recherche IPSL-Climate Graduate School, funding ANR entitled: Programme des Investissements d'Avenir (reference ANR-11-IDEX-0004-17-EURE-0006). R.E.H. is supported by a postdoctoral fellowship from the CNES.

**Institutional Review Board Statement:** Not applicable.

**Informed Consent Statement:** Not applicable.

**Data Availability Statement:** The MAREDAT and Tara Oceans Expedition HPLC data used in this study can be found at https://doi.pangaea.de/10.1594/PANGAEA.793246 (accessed on 1 March 2020). The different merged satellite ocean color data were obtained from the GlobColour project portal (www.globcolour.info) (accessed on 1 March 2020). All the globcolour products are described in the product user guide version version 4.2.1 (https://www.globcolour.info/CDR_Docs/GlobCOLOUR_PUG.pdf (accessed on 1 March 2020)) found on the GlobColour portal. Pathfinder Level 3 Daily Daytime SST Version 5.3 data set were obtained from http://doi:10.7289/V52J68XX/ (accessed on 1 March 2020). Following best practices, the code was deposited into a public domain repository accessible at https://github.com/AgathePuissant/SOM_PCA (accessed on 1 March 2020). Prerequisite software library SOM Toolbox 2.0 for Matlab is required, implementing the self-organizing map algorithm, Copyright (C) 1999 by Esa Alhoniemi, Johan Himberg, Jukka Parviainen, and Juha Vesanto and accessible at https://github.com/ilarinieminen/SOMToolbox(accessed on 1 March 2020).

**Conflicts of Interest:** The authors declare no conflict of interest.

## Abbreviations

The following abbreviations are used in this manuscript:

| | |
|---|---|
| AVHRR | Advanced Very High-Resolution Radiometer |
| Chla | Chlorophyll-A |
| Chla_sat | Chlorophylle-A Satellite measured |
| DVChla | Divinyl Chlorophyll-A |
| ESA | European Space Agency |
| fucox | fucoxanthin |
| HPLC | High Performance Liquid Chromatography |
| ITCOMP-SOM | Iterative Completion Self Organizing Map |
| KDPAR | coefficient of attenuation of photosynthesis available radiance |
| KD490 | light coefficient of attenuation at 490 nm |
| MERIS | Medium Resolution Imaging Spectrometer |
| MODIS | Moderate Resolution Imaging Spectroradiometer |
| NOAA | National Oceanic and Atmospheric Administration |
| OLCI | Ocean and Land Colour Instrument |
| PCA | Principal Component Analysis |
| PAR | Photosynthesis available radiance |
| perid | peridinin |
| PFTs | Phytoplankton Functional Types |
| PSC | Phytoplankton Size Classes |
| RRS412 | Remote Sensing Reflectance at 412 nm |
| RRS443 | Remote Sensing Reflectance at 443 nm |
| RRS490 | Remote Sensing Reflectance at 490 nm |
| RRS555 | Remote Sensing Reflectance at 555 nm |
| SOM | Self Organizing Maps |
| SST | Sea Surface Temperature |
| VIIRS | Visible Infrared Imaging Radiometer Suite |
| zeax | zeaxanthin |
| ZEU | Depth of the euphotic layer |
| ZHL | Depth of the warmed layer |
| 19hex | 19'hexanoyloxyfucoxanthin |

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
