# Peer review of "Inversion of Phytoplankton Pigment Vertical Profiles from Satellite Data Using Machine Learning"

_remotesensing, doi:10.3390/rs13081445_

Round 1

Reviewer 1 Report

Satellite observation is kind of the only way that help monitor the earth surface processes in large scale, but one shortcoming of this technique is the data could only directly reflect any variability at the surface, i.e. for ocean color, only the top 5m could be directly observed! -Regardless of the noise by the difficulties in atmospheric correction.  Therefore, it looks quite exciting to read this manuscript where the authors use the advanced machine learning techniques to model vertical parameters in the ocean. Before being accepted, I have a few minor comments.

1 Basically, this paper modeled two different datasets. One is for the missing observations in explanatory dataset. The other is for the establishment of six photosynthetic pigment profile with six sets of satellite retrievals. But for both models, additional to R2 and RMSE analysis that are already finished, the residue analysis is missing, so as for model diagonosis, in terms of common variance test, un-correlation test, and normality test (as the data reflects variations in nature).

2 In Lines 108-110, is this 9 point logarithmic depth grid from any reference? If so, please add the cited literature. If not, could you give a bit explanation why these 9 depth points are selected? 

3 In Line 131-133, I guess the algorithm comes from the satellite's ATBD documents. Could you cite them here?

4 In Line 222-224, I would suggest moving this narration to Line 140 where you firstly introduced these six satellite parameters, as I initially confused on how these parameters had been selected till the necessary references were given in Line 222-224. 

Reviewer 2 Report

The manuscript entitled “Inversion of phytoplankton pigment vertical profiles from satellite data using machine learning” by Agathe Puissant, Roy El Hourany, Anastase Alexandre Charantonis, Chris Bowler, and Sylvie Thiria proposes a machine learning based method for the assessment of the vertical distribution of phytoplankton pigments from surface satellite observations. This manuscript was previously reviewed by other colleagues and the resubmitted version seems to be improved based on previous comments from other reviewers.

One important remark is that the manuscript is not following the current template for the submission for the journal. Therefore, authors should adequate the manuscript according to the new template available in the journal website (https://www.mdpi.com/journal/remotesensing/instructions).

The second remark is the validation of the satellite data. Authors mentioned that they got the satellite products from the Globcolour project, however it is missing the version of the product that they are using. Additionally, it would be important to understand how much the satellite global products are performing for your global dataset. It is known that global products do not have the same performance for different regions, therefore, it would be important to understand the variability of the remote sensing data before the training of the machine learning. Thus, it would be good to show the relationship between the measure (in situ) and observed (from satellite imagery) Remote Sensing Reflectance in different spectral bands as well as the chlorophyll-a concentration (the High-Performance Liquid Chromatography measurement of chlorophyll-a concentration vs the Globcolour chlorophyll-a concentration product). Just adding a scatterplot of these measurements would show if the satellite imagery are consistent with the measured chlorophyll-a concentration, and if there are corrections or normalization needed before going through the machine learning training.

The third remark, please provide the Root Mean Square Error in %, otherwise it is difficult to access the performance of the method, especially when we do not know the full range of the values. For example, a Root Mean Square Error of 0.05 is low, but if the highest value is 0.1, then the low Root Mean Square Error is not relevant. So, please add the RMSE in % or readers have to look to the mean values column and calculate the percentage to really have an idea of the performance in the Table 3 and in the Table 4.

The fourth remark is on the discussion section which is too short and is missing the discussion of your results as well as the discussion about the uncertainties. Most of the first paragraph was more a literature review for an introduction section than a discussion. I would suggest authors to add more discussion, maybe having more subsections and discuss the results and the uncertainties of the proposed method. Moreover, since it is a proposal of new method, it is important to compare to the current methods and in the discussion you can show the advantages and disadvantages of the proposed method.

Reviewer 3 Report

Overall, authors corrected most of ambiguities in the original manuscript. I appreciate their clear replies and corrections after the first review. However, there are some points to be clarify and minor editorial issues, which are listed below. After the corrections are made, I recommend publication of this study.

REP_REVIEWERS_ARTICLE_AGATHE:

  1. "--The PCA application was intended to reduce...": Table of RMSE btwn smoothed and non-smoothed profiles (supplementary material) and Table 3 do not justify why naxes_i=2 for all pigment gives the best result. It only shows RMSE and correlation scores improves for the smoothed profile with naxes=2. Do you observe monotonic improvement of the skill scores towards naxes=2? Otherwise how do you judge naxes=2 is the best one?

Comment: Assuming that naxes=2 is indeed the best choice, my guess is that key is in the choice of the vertical standard depths. Due to choice of the logarithmic function, vertical resolution near the bottom of euphotic layer is quite coarse (107.81, 179.85,300.0). This is helping MAREDAT observation to be binned in a way favoring naxes=2. So I assume if you choose original MAREDAT depths which is the same as the WOD standard depth, you may find different optimal naxes.

  1. "--The depth of the Pigment concentration .. ". Thank you for this statement and subsequent explanation and Supplementary Figure. As far as I see, this statement is not part of the revised manuscript. Please blend this information to a body of the revised manuscript.

  1. "However, the inter-pigment relationships are". Thank you for this statement and subsequent explanation and Supplementary Figure. The SOM map of classes with 6 pigment + 2 vertical modes shows clear clustering of geographical coordinates which demonstrates that inter-pigment relationships introduces geographical information to SOM. Again, this statement is not blended into the revised manuscript. I recommend insert the statement: "the inter-pigment relationships are specific to regions and to trophic states (Morel and Berthon, 1989), and the variability of these pigments is capable to reflect the influence of environmental factors such as nutrient dependency and water masses on the phytoplankton community (Kheireddine et al., 2017; Pearman et al., 2017). " to adequate place.

REVISED MANUSCRIPT:

L255: [1...9] > [1...6].  I think index i is for pigment, but not for vertical levels.

Round 2

Reviewer 2 Report

Dear authors,

Thank you for providing the response to my comment.

Based on your response, the satellite chl-a and HPLC chl-a got a R2 of 0.62. Chl-a is one of the most detectable pigments that we can measure from remote sensing and 62% of the samples could be explained by remote sensing, while 38% was not explained. Considering that the method that you are proposing the input data is the satellite derived chl-a, this approach starts with 38% of error. Which is the error of the satellite and HPLC (excluding the errors from the HPLC itself). However, this error could be minimized by the use of band ratio of Rrs (which minimizes the effect of light field). When using the Rrs product (without the band ratio), you would have the issues on the light pathway, this is why it is important to evaluate the accuracy of the Rrs.

Then, in your modeling you use Rrs to compute and cannot provide any validation that the Rrs from satellite are accurate, which we know that atmospheric correction is a problem in the field of Remote Sensing of Water Quality.

Because of these issues it is hard to trust the that the results presented in this manuscript. Since the input data for your modeling is not validated.

Author Response

This manuscript is a resubmission of an earlier submission. The following is a list of the peer review reports and author responses from that submission.

Round 1

Reviewer 1 Report

Overall, presentation is poor through the manuscript. Reader needs to go back and forth to understand what has been done and what is the advantage of this proposed method against past studies. I happened to have an experience in retrieving Chl-a profile using the similar approach (SOM), so I can guess what authors did. But if I did not have such experience, it is difficult to understand how the system was constructed and optimized mainly because method and result are not clearly separated in the article. Especially, optimization of super parameters is difficult to follow. Thus, my conclusion of major revision is mainly due to this presentation issue. Scientifically there are many questions as written in the comments to the author about achievement of this study, but approach itself is interesting and this work should be published once the issues are resolved. 

Scientific comments and questions.

P3 L123-L126 "Finally ...  11 satellite variables": This sentence does not make sense grammatically and needs further clarification. What I guess is: D is a matrix of dimension 1164x65. ith column of D, Zi is a vector of dimension 65 consisting of 54 (=9x6) in-situ variables and 11 satellite derived variables.

P4 Figure 1: Transect of the Biosope and the sample ocean point (47N, 8W) should be marked on this map.

P4 Table 1: Presentation and meaning of missing data in this table is not clear. For example, diagram in figure 2 indicates satellite data (1614 x 11) at the training stage does not contain missing values. Does it mean the number of collocated profiles, 1614, are derived only at the satellite grid with non-missing value?

P5 eq.1: In page 3, Z_i is a vector of dimension 65 (=11+54), but length of referent vector W_c is (11+n_axes) according to Figure 2. Further, Z_i contains missing data in this study. My understanding on eq.(1) is that it is introduced here for explaining SOM in more general sense, but using Z_i for slightly different meanings in the same article makes reader confused if eq.(1) needs to be referred later.  

P5 L154: What is the meaning of  "m (dimension of training data D)"? m should be a dimension of a set of in-situ data.

P5 L164-166 "Moreover, .. in the case of missing data": This sentence is difficult to understand for reader not familiar with ITCOMPSOM. Is it inferring that the method use multivariate correlation information to fill the missing value? Please explain more detail if it is important to this study or simplified to reduce its ambiguity.

P6 Figure 2. Honeycomb map in the center of this figure is SOM topological map, but there is no explanation in main body of paper about if the same SOM topological structure used for ITCOMPSOM, which is described in subsection 3.1, is used for the training. Needs clarification.  

P6 L183-: What is the SOM topological structure during this training phase?. Is the same SOM topological structure used for ITCOMPSOM, which is described in subsection 3.1, is used for the training too?. Then please explain it in the paragraph following below.  

P6 L184-186 "The nine .. PCA": This sentence does not make sense with me. What do you mean by "the nine principle coordinates"?. My understanding based on the explanation in subsection 3.1 is the PCA is performed on a covariance matrix dE^{T}dE/1613 (T is for transpose) of dimension (9x9), where E is a matrix of in-situ data for each pigment of dimension (1614 x 9) and dE=E-<E>. Is it right? Needs further clarification.

P6 L191-196: observed data is 11 satellite-derived variables and missing data is n_axes (and 54 after re-projection to physical space) in-situ data. This should be mentioned clearly within in this paragraph.

P8 L259-L260: Keeping only two PCA means vertical profile of each pigment likely to mean the profile can be explained by two modes: (i) uniform pigment within euphotic layer and below and (ii) depth change of the euphotic layer. In order to explain the sub-surface maximum retrieved in figure 3, it may require at least one more vertical mode. It would be helpful to plot shape of the two leading PCAs (vertical profiles).

P8 L260-L262: It is explained in subsection 2.2.4 that the choice of the six satellite-derived variables is made based on the findings of [9], but it is mentioned here it is based on the cross-validation. These two explanations does not look compatible. Needs further clarification.

P9 L296-298 and P10 Fig.3(c): Color shade on figure 3c is hard to recognise. Can you change the range of colorbar to such as -1 to 1?

P10 L300-: Which year of satellite data did you use for the sub-surface reconstruction? When does the series of weeks start and end?

P11 L322-L323 "This analysis .. region": This statement is too vague. Does it means "diatom-dominated" spring bloom and its timing is supported by past studies? Does the past studies also support difference in the peak depths, 23m-38m for Fucox and 8.35m for Zeax seen in the retrieved vertical profiles? This is important to be verified since you mentioned that this new method is useful for "robust estimations of the vertical variability".

P11 Figure 5. It is strange to see that the depth of uniform Chl-a layer is almost constant at 64.63m through the entire year. My suspicion is on the choice of keeping only 2 vertical modes in PCA. You should discuss about this point in section 4.

P12 L349 "with a resolution in the range of kilometre": This sentence is not clear. Are you talking about vertical resolution or horizontal resolution?

P12 Figure 7: Highest value of the retrieved Zeaxanthin is at week 1-2. Why?

P12 L338-355: Uitz et al (2006) also requires geographical data as input to estimate vertical Chl-a profile. In general, phytoplankton composition or functional types (FTPs) depends on not only physical condition (light and temperature) and stage of bloom which are inferred from the satellite data, but also depends on nutrient availability which highly depends on water mass properties. Geographical data are necessary in these past studies aiming at deriving FTPs at surface and subsurface for compensating lack of the water mass information in satellite-derived variable. In this new method proposed here, I do not see a reason why this method can over come this problem. Author should discuss this clear way in the discussion.

Editorial corrections:

P2-L50:  Physical and biogeochemical > physical and biogeochemical

P2-L55: Satellite ocean colour daily Matchups > satellite ocean colour daily matchups

P5 equation (1):  the Zi in the bracket on the right hand side should be simply Z.

Reviewer 2 Report

Review of “Inversion of phytoplankton pigment vertical profiles from satellite data using machine learning”

Presented is an inversion of phytoplankton pigment vertical profiles from satellite data using the Self-Organizing Map (SOM). The topic is interesting.  Although very challenging, some encouraging results have been presented. The SOM is a powerful technique that can be used for this kind of applications. I would like to recommend the manuscript be accepted for publication after some necessary clarifications and addition of relevant publications.  Specific comments are listed as follows:

  • Figures 3b & 4b, the black areas correspond to the longitudes not having enough matched satellite data available to perform the inversion (Lines 288 – 289). How do you define the criterion -- enough or not enough matched satellite data available to perform the inversion? This is critical in this study. Be specific here. It is mentioned 50% on Line 92, but that is about pigment observations, not satellite data.

  • Line 169 – 173, in your PCA filtering, how many modes are used in the reconstruction of the data? Any justifications of the choice of the leading modes? This info is missing from the text.

  • Line 356, the method is globally applicable, why “excluding the South Ocean”? According to Fig. 1, no in situ data in the Southern Ocean. However, not much data in the western Pacific, western Atlantic and eastern Indian Ocean, either.

  • Line 146, SOM application in meteorology and oceanography, a relevant publication should be cited here: (editor remove references)

  • Line 147, it would be good to add the following to the end of the paragraph: “The SOM is demonstrated to have better performance than conventional techniques in identifying characteristic patterns (references removed by editor). Also, SOM has been to extract the relationship between surface and subsurface patterns in the ocean (references removed by editor).”

Reviewer 3 Report

The technical note entitled "Inversion of phytoplankton pigment vertical profiles from satellite data using machine learning" by Agathe Puissant, Roy El Hourany, Anastase Alexandre Charantonis, Chris Bowler, and Sylvie Thiria proposes a machine learning methodology to infer the vertical distribution of phytoplankton pigments based on the surface estimation from satellite observations. With this procedure authors tried to provide new insights of the vertical dynamic of phytoplankton in the water column  with a high spatial and temporal resolution.

The major problem of this study is the poor discussion. Authors should discuss the uncertainties of the proposed approach because there is a sum of different errors, the error of the ITCOMPSOM, then the error of the PCA, the error of the Globcolour project products, and others. Therefore, it is crucial to discuss the uncertainties of the proposed framework. Additionally, authors should compare their results to other studies to show the advantages of the model.

Another issue regards the training dataset, maybe authors could add more details about it so readers will have more confidence in your results. It would also be realistic to present the limitations of the methodology, for example, the method can only work for certain optical water types. Overall the manuscript needs a careful revision because it cannot convince readers that the proposed approach really works.

Here are some specific comments to help you to improve it.

Specific comments:

L3 - please add a little bit more details about the in situ methods, especially to highlight the use of remote sensing.

L8 - please avoid the repetition of the term "Self-Organizing Map"

L11 - it would be better to present the error estimator.

L13 - phytoplankton types maybe is too subjective, I believe it is better to really state which ones you could estimate.

L19 - please add a reference to support this statement.

L30 - cohesion and coherence are missing among sentences, the paragraph reads like a collection of bullet points. Authors should improve the link between two sentences.

L33 - it will be difficult to infer the deep chla maximum if the surface data

L48 - please use author's name not the journal.

L54 - add the acronym for HPLC and for MAREDAT

L65 - are these data collected at the same time?

L81 - please add that you will explain how the "quality control" works.

L82 - mg/m³

L83 - does it mean that phytoplankton bloom is excluded? 

L85 - what are the optical problems for the Arctic waters?

L98 - which ones?

L122 - please add the number of samples before merging satellite and in situ.

L155 - please explain the acronym.

L157 - maybe it would be good to evaluate the performance of the data filling, especially if 60% of the satellite data for chla is missing.

L207 - it would be better to use a normalized estimator (%)

L244 - what is the distance between stations?

L254 - It would be good to show a validation of the ITCOMPSOM.

L260 - seems that taking 3 components would be better, since for some datasets 2 components explain 69%.

L298 - satisfactory is subjective, it would be better to present a statistical estimator.

L299 - Figure 4, a scatterplot (observed and estimated) would be more efficient.

L311 - please add a reference to support this statement.

L338 - 347 - this is mostly a repetition of the introduction.

L348 - "other studies", authors should add more studies not only 1 (Ref 33).

L356 - while was it excluded? Due to lack of data or because the quality control?